# Trust Is for the Strong: How Health Status May Influence Generalized and Personalized Trust

**DOI:** 10.3390/healthcare11172373

**Published:** 2023-08-23

**Authors:** Quan-Hoang Vuong, Phuong-Loan Nguyen, Ruining Jin, Minh-Hoang Nguyen, Tam-Tri Le

**Affiliations:** 1Centre for Interdisciplinary Social Research, Phenikaa University, Hanoi 100803, Vietnam; hoang.vuongquan@phenikaa-uni.edu.vn (Q.-H.V.); hoang.nguyenminh@phenikaa-uni.edu.vn (M.-H.N.); 2Spanish Department, Hanoi University, Hanoi 100000, Vietnam; loan.nguyen@hanu.edu.vn; 3Civil, Commercial and Economic Law School, China University of Political Science and Law, Beijing 100088, China; 4A.I. for Social Data Lab (AISDL), Vuong & Associates, Hanoi 100000, Vietnam

**Keywords:** generalized trust, personalized trust, health condition, information processing, Bayesian Mindsponge Framework

## Abstract

In the trust–health relationship, how trusting other people in society may promote good health is a topic often examined. However, the other direction of influence—how health may affect trust—has not been well explored. In order to investigate this possible effect, we employed the Bayesian Mindsponge Framework (BMF) analytics to go deeper into the information processing mechanisms underlying the expressions of trust. Conducting a Bayesian analysis on a dataset of 1237 residents from Cali, Colombia, we found that general health status is positively associated with generalized trust, but recent experiences of illnesses/injuries have a negative moderating effect. Personalized trust is largely unchanged across different general health conditions, but the trust level becomes higher with recent experiences of illnesses/injuries. Psychophysiological mechanisms of increasing information filtering intensity toward unfamiliar sources during a vulnerable state of health is a plausible explanation of found patterns in generalized trust. Because established personal relationships are reinforced information channels, personalized trust is not affected as much. Rather, the results suggest that people may rely even more on loved ones when they are in bad health conditions. This exploratory study shows that the trust–health relationship can be examined from a different angle that may provide new insights.

## 1. Introduction

Trust is a crucial aspect of human society. The expressions of trust in human behavior are complex and heavily context-dependent. Studying its patterns may require a deeper understanding of the underlying mechanisms in terms of information processing. Considering the notion of trust in a broader scope may provide some interesting ideas about how to approach such processes. As an example, people sometimes observe that their pets may act a bit more cautiously towards strangers and new things when they are sick or injured but retain a relatively stable level of trust towards their owner across different health conditions. Such intuitive observations may suggest that a discrepancy stems from different trust types towards other parties. It hints at some questions about possible psychophysiological pathways underneath the expression of trust beyond the boundary of interpersonal trust. In the context of human interpersonal interactions, generalized trust (general trust or social trust) is trust toward other people in society in general, including strangers, whereas personalized trust (personal trust or particularized trust) is trust toward people with already established relationships [1]. The present study is an exploratory study aiming to investigate more deeply the relationship between health and trust based on principles of information processing. Specifically, we examine how health status may affect generalized trust and personalized trust differently.

### 1.1. The Current Landscape of Human Trust Research

Research on interpersonal relationships treats the concept of trust in many different scopes depending on the context of examination. Some regard trust as the willingness to be vulnerable to the trusted [2,3]. From a more analytic view, trust is seen as a chronological mental process involving three elements of expectation (process outcome), interpretation (rational and emotional evaluation), and suspension (moderating interpretative knowledge) [4]. The duality of control and trust based on the interpretation–suspension interactions on expectation [5] is in alignment with the influencing–influenced dynamic in the subjective sphere formation and optimization processes [6]. In general, trust is often considered an essential property that helps keep the stability of social structures [7]. However, trust is a complex psychological concept and may not always be characterized by a trustor’s willingness to be vulnerable, dependence on the trustee’s goodwill and competence, and anticipation of a favorable response from the trustee [8,9]. When considering decision-making, trust is the result of a rationalization process [10,11], meaning that it is formed through information filtering [12].

Regarding the link between health and trust, studies found that trust promotes better physical and mental health [13,14]. However, this relationship is not straightforward. Some studies suggest that the positive association between generalized trust and health outcomes is moderated by the human development index, in which countries/societies with relatively higher levels of development have a stronger relationship [15,16]. Other studies found that different types of trust have different effects on people’s health in different living environments with inconsistent patterns [17,18]. Regarding the aspect of social condition, it is also suggested that in a low-trust society, high individual-level generalized trust can be positively associated with depressive symptoms [19]. Nevertheless, extant literature mainly focuses on examining how trust may affect health. Interpretation of the other possible direction of influence in the relationship has not been well-explored. A more recent study on politics reveals that people in poor health tend to have lower levels of trust in the political system compared to people in good health [20]. However, the reasoning for this type of influence is mostly presented in terms of political preferences or expectations of public services in the context of a relatively developed society in Europe [20,21].

### 1.2. Hints from Observations in Nature on the Health-to-Trust Direction

In order to explore the possible direction of health influencing trust, it is necessary to look at the aspect of psychophysiology due to the working of the human mind and body. In the natural world, biological organisms need to balance between information exchange with the external environment and keeping the homeostasis of their own systems. Even plants rely on information processing mechanisms for decision-making to flexibly orchestrate internal growth priority and responses to external stimuli [22]. Animals show complicated and dynamic risk–resource trade-offs in strategizing the survival of individuals as well as the long-term growth of the population [23]. Noteworthily, on the basis of neuronal plasticity, injuries can affect nociceptive sensitization (involving hyperexcitability of neurons following injury to their axons) and change the injured organisms’ behaviors accordingly [24]. Interestingly, studies in squids found that injuries enhance nociceptive sensitization, which leads to hyper-responsiveness toward predators and helps reduce predation risks [25,26]. Adaptive attitudes and behaviors based on information processing in biological systems, regardless of complexity levels, are fundamentally for the sake of survival [27,28].

### 1.3. An Information-Processing Approach for Examining the Trust–Health Relationship

Effectively investigating the trust–health relationship in this approach requires a compatible conceptual framework that goes deeper into the underlying information-processing mechanism of the mind. In the present study, we use Bayesian Mindsponge Framework (BMF) analytics [29,30]. A detailed rationale for applying BMF analytics and constructing models is presented in the Methodology section. Furthermore, patterns of survival-driven information processes are likely stronger in relatively more unstable social conditions. Here, we will examine patterns of trust in the high-violence context of Colombian urban society [31]. To find differences in adaptive responses between non-reinforced and reinforced trust, the influence of health status on generalized trust and personalized trust will be compared. Additionally, to find supporting evidence for the patterns in terms of information density, the factor of recent experiences of illnesses/injuries will be tested for possible moderating effects. The research questions (RQs) in the present study are as follows.

RQ1: Is there an association between general health status and generalized trust?

RQ2: How do experiences of recent illnesses/injuries affect the relationship in RQ1?

RQ3: Is there an association between general health status and personalized trust?

RQ4: How do experiences of recent illnesses/injuries affect the relationship in RQ3?

## 2. Methodology

### 2.1. Theoretical Foundation

#### 2.1.1. Mindsponge Theory Overview and the Information Multi-Filtering System

Quan-Hoang Vuong and Nancy K. Napier created the term mindsponge in their early studies on acculturation and globalization [32]. The notion was described as a dynamic process of how a mind assimilates new cultural values and discards waning ones in response to environmental conditions. Mindsponge was expanded into a theory of how the mind processes information [27]. The mind, according to the mindsponge theory, is a collection-cum-processor of information. The functions of information storage and information transforming/filtering create a feedback loop that enables dynamic collecting and processing activities in an updating manner. This broad definition of the mind includes biological and social systems on individual and collective levels, where the boundaries of an examined system depend on the observer’s perceptions. The complexity level of a system can range from simple nervous systems to advanced human minds and collective mindsets of societies [27]. The extended mindsponge theory was established using the latest findings from brain and life sciences, taking into account the fundamental physiological structures and activities of humans.

There are some basic components and activities of the mind. The mindset is the collection of approved information stored in the system’s memory. The filtering mechanism regulates what information enters or leaves the mindset based on the content of the current mindset. The act of information filtering affects both the mindset and the filtering mechanism. If necessary, the trust mechanism (selective prioritizing) can be used to expedite the filtering procedure and conserve energy. A mindsponge information process has the following characteristics:It depicts the fundamental patterns of the biosphere system.The procedure is one that is both dynamic and balanced.It uses a cost-benefit analysis and aims to maximize perceived benefits while minimizing perceived costs for the overall system.Because it uses energy, it follows the principle of energy conservation.It adheres to objectives and priorities based on the needs of the system.Its major function is to sustain the continued existence of the system, as manifested by survival, growth, and reproduction.

The updating mechanisms in human minds are highly flexible “live-wiring” thanks to neuroplasticity—the ability of neurons and neural networks in the brain to change their connections and behavior in response to new information, sensory stimulation, development, damage, or dysfunction [33]. These biochemical changes in neurons and neural networks reflect the abstract notion of information processing. For example, dynamic information storage and connection generation are enabled by engrams (cognitive information imprinted in physical substances) and long-term potentiation (strengthening activated synapses) [34,35,36]. The updating mechanisms in human minds are highly flexible “live-wiring” thanks to neuroplasticity as opposed to the popular “hard-wiring” method in simpler systems (relying more on genetic information and instincts) [37,38]. Information absorbed from the environment and incorporated into a person’s mindset is kept in the form of trusted values [29]. Stored values are dynamically modified to respond to a changing external environment based on related experiences, such as newly obtained information and newly conceived ideas [39]. The system optimizes itself over time to better align mental representations (subjective values) with reality (objective values) [6]. Bayes’ Theorem (shown below) offers a helpful mathematical basis for understanding the information process of updating beliefs.
(1)$$p\left(\theta |X\right)\propto L(X|\theta )p\left(\theta \right)$$

The theorem can be interpreted as the posterior probability distribution $p\left(\theta |X\right)$ being proportional to the prior probability distribution $p\left(\theta \right)$ and the likelihood function L(X|θ). This mathematical concept has two derived notions in the context of the human mind—the current state of mind is based on continuous related past mental processes, and future processes are based on the current content. Consequently, these have implications for statistical analysis; cross-sectional data represent the observable results of related past information processes, and the estimated posterior of the present study can be used as the prior in future studies to update the findings, adapting the discovered patterns in light of new evidence. The equivalence between specific evolutionary dynamics and Bayesian inference provides insight into the evolution of human cognition [40]. Due to the similarity in information processing principles between Bayesian inference and human cognition (such as dealing with uncertainty due to incomplete information, the updating manner, etc.), Bayesian modeling is a useful tool for studying the operation of the human mind [41,42].

Changes in content in the mindset are driven by the assimilation of new information considered beneficial and the rejection of old information deemed no longer acceptable. As a result of mindset shifting, new values are filtered differently, which changes how the entire system works.

#### 2.1.2. The Trust Mechanism in the Mindsponge Framework

Information processing systems can prioritize certain inputs for easier processing while prohibiting others from entering the processes in order to conserve energy. A system’s entropy rises as it absorbs more information from its surroundings. To retain its highly organized structure, a biological system must expend energy (Schrödinger, 1992). All biological systems are constrained by their available energy capacity, and organisms have continuously evolved to maximize their energy efficiency in response to the intense competition for resources.

Various energy-saving systems may be found everywhere in nature, especially in biological organisms [43]. The issue of energy constraint is especially critical when it comes to the human brain, which accounts for only 2% of body mass yet uses approximately 20% of the total oxygen and calorie intake [44]. At the stage of rapid development in small children, the brain may consume up to fifty percent of the body’s energy [45]. A baby’s brain has roughly 10^14^ synapses, which are reduced to half that amount during maturation when redundant synapses are removed in favor of strengthening well-used circuits [37]. Furthermore, it is suggested that the neuronal wiring of the neocortex of the human brain evolved so that it can accomplish complicated operations with minimal energy expenditure [46]. Sparse coding allows the brain to “compress” information to save energy, making processing more efficient [47].

For the human mind and society, trust can be considered a sophisticated selective information channel. By acting as a “priority pass”, trust accelerates the filtering procedure. Given that an individual must filter a huge volume of information from their infosphere, the trust evaluator’s “priority pass” method is often utilized to improve the efficiency of information processing. Casually speaking, nobody can meticulously investigate everything they encounter. The gradient of trust degrees covers distrust, indifference, and trust. The mind can conserve more energy and time by using trust to swiftly assign values to related information rather than analyzing it from the beginning using rigorous cost–benefit assessments. Before a value can be utilized as a reference for trust, it must first be accepted and incorporated into the mindset in order to generate reliability for similar values. In cases of mistrust, the integrated value fosters unfavorable responses, such as avoidance or hasty rejection of similar values. In addition, the burden of one’s own filtering system can be decreased by assuming that information coming from a trustworthy source has been evaluated by the source’s filtering system.

When information moves from the environment into the mind, trust evaluators serve as gatekeepers [32,48]. As new information comes in, the trust evaluator is engaged to determine whether the present mindset already contains relevant prior information. Without established trust (or distrust), new information is subject to a thorough cost–benefit analysis before its values can be accepted or rejected. Depending on the corresponding value of trust, the information is attached with positive values (trusted) or negative values (distrusted). In brief, trust reduces the intensity of the cost–benefit analysis and raises the net value of the information being assessed, hence increasing the likelihood of acceptance. In circumstances of extreme distrust, the large additional negative values may result in quick rejection without the usual evaluation process. Alternatively, the person may opt to deliberately avoid obtaining information from distrusted sources, which is analogous to information inaccessibility. This trust mechanism affects how people carry out information-seeking behavior and interpretation of such information, which is likely the underlying reason for the confirmation bias phenomenon [49]. The mindsponge theory and BMF analytics have been effectively employed to study the role of the trust mechanism in human subjective perceptions [48,50,51].

### 2.2. Model Construction

#### 2.2.1. Materials and Variable Selection

Because the mindsponge mechanism provides an analytical framework that aids in the construction and simulation of a psychological process using the data at hand, BMF analytics are compatible with survey data [52]. Binary variables (reflecting information availability, objective condition, etc.) and continuous variables (e.g., reflecting information density, belief strength, etc.) are the two most prevalent types of variables used in BMF analytics. Technically, ordinal and discrete variables can be treated as continuous variables [53]. Regarding data usage, we adhered to the FAIR principles for scientific data management and stewardship: Findable, Accessible, Interoperable, and Reusable [54].

Using secondary data from the dataset of Martínez [55], the current study examined the associations between the level of trust (generalized trust and personalized trust) and people’s health in Cali, one of the major cities in Colombia, South America. The dataset was collected through face-to-face surveys designed by the Observatorio of Políticas Públicas–POLIS—of Universidad Icesi’s trained pollsters in 2017. The survey follows international guidelines and includes three questions about health adapted from the Centers for Disease Control and Prevention. The interpersonal and institutional trust component was based on OECD guidelines to measure trust.

Before conducting the survey, the questionnaire was piloted 20 times for language adjustment corresponding to the local context. The dataset is a representative sample of the adult population in the city. The sample size was estimated using population reports of the National Statics Office in Colombia. The survey was designed with three stages: (1) selection of 38 points around the city; (2) defining quotas according to socioeconomic strata, gender, race/ethnicity; (3) random selection of target population. The survey was delivered by pollsters in multiple locations selected in the city. Pollsters approached respondents, explained the purpose of the study, and assured confidentiality. After their voluntary participation in the project, respondents received a bookmark with the project information. Each survey was revised by field supervisors. The data log was then recorded day by day to control the ratio set about the demographical data of representatives. A total of 1237 responses were collected. More detailed descriptions of the dataset can be found in the open-access data article by Martínez [55] (https://doi.org/10.1016/j.dib.2019.104639) (accessed on 13 March 2023).

Four variables are employed for Bayesian analysis (see Table 1) based on the conceptual models presented in the Model Formulation subsection.

The variable *GenTrust* comes from the question, “In general, how much do you trust most people?”. The variable *PerTrust* comes from the question, “In general, how much do you trust most people you know personally?”. The variable *Health* comes from the question, “Would you say that, in general, your health is […]”. Note that the direction of value order (poor to excellent) is flipped compared to the original questionnaire. The variable *Recentillness* comes from the question, “Now thinking about your physical health, which includes physical illness and injury, for how many days during the past 30 days was your physical health not good?”. All values but zero are coded as 1.

#### 2.2.2. Model Formulation

The models for statistical analysis are constructed based on the theoretical foundation presented above. Trust is a mental mechanism used to facilitate information reception and filtering, which reduces the energy expenditure of the filtering process. It should be noted that this mechanism is ultimately employed for the sake of the survival and functioning of the system. Thus, when a system is in a vulnerable state, it is logical that the intensity of information-seeking activity is lessened in favor of strengthening the trust guards to protect oneself against potential harm. This trade-off is necessary because self-preservation is naturally prioritized in biological systems, especially when there is damage (such as illnesses and injuries). It is not intuitive to facilitate information exchange with unfamiliar interactants when the system is currently more prone to risk. Here, trust guards (or “gatekeepers”) may need to be stricter than normal until the system is finished repairing.

In the case of humans, trust toward strangers may be lessened when the body is dealing with physical problems. Thus, it is likely that generalized trust may be lower when one’s general health condition is relatively worse. To support this direction of interpretation, we need to consider the moderating effect of recent experiences of illnesses/injuries. This possible moderating effect makes the interpretation direction of trust-to-health less reasonable due to the chronological nature of recent events. The moderator is expected to intensify the aforementioned potential association between health status and generalized trust. Model 1 is as follows.
(2)$$GenTrust\sim normal\left(\mu ,\sigma \right)$$
(3)$$\mu_{i}=\beta_{0}+\beta_{Health}*{Health}_{i}+\beta_{Recentillness*Health}*{{Recentillness}_{i}*Health}_{i}$$
(4)$$\beta \sim normal\left(M,S\right)$$

Figure 1 shows the visualization of Model 1’s logical network where generalized trust *GenTrust* is possibly predicted by individual i’s health condition of ${Health}_{i}$. Such an association is moderated by individual i’s recently experienced illnesses/injuries ${Recentillness}_{i}$. The model has intercept $\beta_{0}$ and coefficients $\beta_{Health}$ and $\beta_{Recentillness*Health}$.

While trust toward strangers may be influenced by one’s physical health, we hypothesize that trust toward those with established relationships is not influenced by one’s physical health. Personalized trust is a reinforced value filtered through multiple prior interactions. Furthermore, available related information that can be served as references for evaluating established personal relationships should be of higher density compared to non-reinforced relationships. Thus, it is likely that personalized trust is less influenced by one’s bad health condition. Here, we also need to consider the moderation of recent illness experiences to support how the association is interpreted. Unlike Model 1, *Recentillness ∗ Health* in Model 2 is expected to not have a positive association with *PerTrust*. Model 2 is as follows.
(5)$$PerTrust\sim normal\left(\mu ,\sigma \right)$$
(6)$$\mu_{i}=\beta_{0}+\beta_{Health}*{Health}_{i}+\beta_{Recentillness*Health}*{{Recentillness}_{i}*Health}_{i}$$
(7)$$\beta \sim normal\left(M,S\right)$$

Figure 2 shows the visualization of Model 2’s logical network where personalized trust *PerTrust* is possibly predicted by individual i’s health condition of ${Health}_{i}$. Similar to Model 1, this association is moderated by individual i’s recent illnesses/injuries experienced as ${Recentillness}_{i}$. The model has the intercept $\beta_{0}$ and coefficients $\beta_{Health}$ and $\beta_{Recentillness*Health}$.

The procedure for conducting Bayesian analysis on the models is presented in the following subsection.

### 2.3. Analysis and Validation

Following the protocol of BMF analytics, our study uses Bayesian analysis assisted by Markov Chain Monte Carlo (MCMC) algorithms [29,30]. The BMF is utilized in this investigation due to its numerous benefits. The mindsponge mechanism and Bayesian inference are highly compatible, both in terms of philosophical approach and technicality. Bayesian inference evaluates all properties probabilistically, allowing for accurate prediction when working with parsimonious models. By leveraging the capabilities of the MCMC methods [56,57], Bayesian analysis can be used with many types of models, such as multi-level correlation structures and non-linear regression frameworks, resulting in a considerable advantage of adaptability. Additionally, in contrast to the frequentist approach, the Bayesian approach interprets reliability using credible intervals rather than the *p*-value, which may involve the risk of overdependence and rigidity.

Regarding validation techniques, The goodness-of-fit of the models is evaluated using Pareto-smoothed importance sampling leave-one-out (PSIS-LOO) diagnostics [58]. LOO is computed as follows.
$$LOO=-2{LPPD}_{loo}=-2\sum_{i=1}^{n}\log\int p\left(y_{i}|\theta \right)p_{post\left(-i\right)}(\theta )d\theta$$

$p_{post\left(-i\right)}(\theta )$ is the posterior distribution based on the data minus data point i. *k*-Pareto values are used in the PSIS method to compute leave-one-out cross-validation in R’s “LOO” package. *k* values help to identify observations with a high degree of influence on the PSIS estimate, which may negatively affect the estimation of the leave-one-out cross-validation. When *k*-Pareto values are greater than 0.7, observations are often considered influential and need to be examined more closely. Normally, a model can be considered to have an acceptable goodness-of-fit when the *k* values are below 0.5.

The convergence of Markov chains can be visually checked using trace plots, Gelman–Rubin–Brooks plots, and autocorrelation plots. It is also statistically checked using the effective sample size (*n_eff*) and the Gelman–Rubin shrink factor (*Rhat*). The *n_eff* value represents the number of non-autocorrelated iterative samples during stochastic simulation. If the *n_eff* values are more than one thousand, the effective samples are sufficient for reliable inference. The *Rhat* value (Gelman shrink factor) is also used to assess the convergence of Markov chains. If the *Rhat* values are above 1.1, the chains likely do not converge. If *Rhat* equals 1, the model can be deemed convergent.

The **bayesvl** R package (Version 0.8.5) [59] was used to conduct Bayesian analysis. It is an open package with good visualization capability and efficient operation [60]. The model’s MCMC configuration comprises 5000 iterations, including 2000 warm-up iterations and four chains. Considering the importance of transparency and the cost of science [61,62], all data and code snippets of this study were deposited at an Open Science Framework server (https://osf.io/czd9t/) (accessed on 13 March 2023).

## 3. Results

The latest model fitting runs were conducted on 4 March 2023 on R version 4.2.1, Windows 11. The total elapsed time was 127.9 s for Model 1 and 105.2 s for Model 2.

### 3.1. Model 1: Generalized Trust

Model 1’s goodness-of-fit was checked using PSIS-LOO diagnostics. As shown in Figure 3, all *k* values are below 0.5, indicating good model specification.

Table 2 shows that, for all parameters, the *n_eff* values are more than 1000, and the *Rhat* values equal 1. These statistical results mean that the Markov chains are well-convergent.

Convergence check can also be conducted visually using the trace plots (Appendix A), the Gelman–Rubin–Brooks plots (Appendix A), and the autocorrelation plots (Appendix A). In the trace plots, the Markov chains fluctuate around central equilibriums. In the Gelman–Rubin–Brooks plots, the shrink factor values rapidly drop to 1 during the warm-up period. In the autocorrelation plots, the average autocorrelation levels are reduced to zero after a finite number of lags. All these indicators suggest good convergence of the Markov chains.

Figure 4 shows that most of the estimated posterior distributions of *Health* are located on the positive side of the *x*-axis, and most of the estimated posterior distributions of *Recentillness ∗ Health* are located on the negative side of the *y*-axis, which indicates that the estimated effects are reliable. Health is positively associated with *GenTrust* ($M_{Health}=0.21$ and ${SD}_{Health}=0.07$). *Recentillness ∗ Health* negatively moderates the relationship between *Health* and *GenTrust* ($M_{Recentillness*Health}=-0.08$ and ${SD}_{Recentillness*Health}=0.04$).

To aid result interpretation, estimated generalized trust values are calculated based on the posterior coefficients of the model. Figure 5 shows the visualization, where the *y*-axis represents the degree of generalized trust, the *x*-axis represents health status, and the line color represents recent illness experience. Here, both lines go up, and the “no recent illness” line is above the other.

### 3.2. Model 2: Personalized Trust

Model 2 also has a healthy goodness-of-fit, as shown through the PSIS-LOO diagnostic plot (Figure 6).

As shown in Table 3, all *n_eff* values are above 1000 and all *Rhat* values equal 1. Additionally, for Model 2, the trace plots (Appendix A), the Gelman–Rubin–Brooks plots (Appendix A), and the autocorrelation plots (Appendix A) all show good signals of convergence.

Figure 7 shows that most of the estimated posterior distributions of *Recentillness ∗ Health* are located on the positive side of the *y*-axis, indicating that the effect is reliable. *Health* does not have a significant influence as its posterior distributions lie around zero ($M_{Health}=0.00$ and ${SD}_{Health}=0.08$). However, *Recentillness ∗ Health* has a positive moderating effect on the above relationship ($M_{Recentillness*Health}=0.09$ and ${SD}_{Recentillness*Health}=0.05$).

Figure 8 visualizes the estimated personalized trust values calculated based on the posterior coefficients of Model 2 in a similar manner to Model 1. Here, the “no recent illness” line (green color) is largely unchanged across the *x*-axis. The “recent illness” line (blue color) is located above the green line and slightly goes up, meaning a positive association (small magnitude) between *Health* and *PerTrust*.

## 4. Discussion

Employing BMF analytics on 1237 Colombian urban residents, we found that general health status is positively associated with generalized trust, but recent experiences of illnesses/injuries have a negative moderating effect. We also found that personalized trust is largely unchanged across different general health conditions, but the trust level becomes higher with recent experiences of illnesses/injuries. The positive association between generalized trust and health status is in alignment with former studies on the relationship [13,14,15,16]. There is a seemingly similar association between poor health and political trust [20], but unlike the present study, that relationship deals with indirect interactions with representative institutional entities. The other patterns found in the results suggest that the relationship is complex and not straightforward, as discussed by some studies [16,17,19]. Particularly, the comparison between the patterns of generalized trust and personalized trust with the moderating effects of recent illness experience points to the possible underlying psychophysiological pathways.

### 4.1. The Trust Mechanism in Relation to One’s Health

In terms of information processing, trust is a mechanism used to facilitate information reception and filtering. In normal conditions, this mechanism is very helpful when interacting with the surrounding infosphere by reducing the energy and time spent on subjective cost–benefit analyses [12,48]. Thus, trust enables fast information absorption and makes social interactions (information exchange) more efficient. However, it would not make sense to prioritize normal information-seeking activities when the body (the physical platform for the mind’s functioning) is in a vulnerable state of health. As a simplified example, an injured/sick person likely is more wary of strangers due to their current low capability of self-defense (physically or psychologically). A high-violence context can make the pattern more clear. For example, someone in a weak state of health will really think twice about walking through a bad neighborhood due to a higher risk of being tricked, assaulted, robbed, etc.

On a neurobiological level, nociceptive information processing through complex neural pathways allows individuals to actively adjust their behaviors to avoid harm, protect the wounded parts, and enhance natural recovery [63]. In the animal kingdom, especially highly evolved organisms like mammals, nociceptive sensitization following injuries can generate persistent changes in the nervous system, which causes exaggerated and prolonged pain to subsequent stimuli and induces avoidance responses [24,64]. This mechanism was found to lower the risk of predation [26]. Regarding more advanced human cognitive processes, the influencing psychophysiological pathways and corresponding behavioral expressions can be even more complex. Nonetheless, a similar pattern can be observed, as presented above. Interestingly, a study found that people in a bad emotional state are more analytical when processing text information [65]. For those who are in a vulnerable state of health, the trust guards are likely engaged in greater intensity to lower the probability of accidentally accepting harmful information. Thus, the level of trust toward strangers is lower in such conditions.

### 4.2. Differences between Generalized and Personalized Trust in Terms of Information Processing

Unlike generalized trust, personalized trust is values reinforced through prior interactions. Intuitively, one does not simply become more cautious toward close family members and friends just because they are in a bad state of health. The notions of spheres of influence and being influenced [6] can be helpful in examining these information processes. For those who have sickness or injuries, the sphere of influence (both objective and subjective) is reduced. In other words, their capabilities of producing active and reactive actions are lower than normal. Thus, perceived impacts from the sphere of being influenced are intensified. Note that this is perceived intensity due to strongly activated trust evaluators and not because the perceivable range is increased [66]. In the case of healthy individuals, new information can be granted quicker and easier entry due to the buffering effect of responding capabilities. In the case of unhealthy individuals, new information likely needs to go through a more rigorous evaluation process. Here, if the inward flow of influence is not strictly controlled, the weakened system may not be able to produce appropriate counteractions in cases of harmful influences. On the other hand, established personal relationships are reinforced information channels. Influences from these sources are already validated to be “safe” and thus do not require intensified reliability checking [12]. Furthermore, the positive moderating effect from our results in Model 2 suggests that illnesses or injuries may make people rely more on their loved ones, indicated by the increased personalized trust. In other words, when being vulnerable, people may give even more “generous” priority passes to information sources that they already trust. Probably, this adjustment helps balance the increased energy expenditure spent on strengthening the filtering processes toward unfamiliar sources.

### 4.3. Recommendations for Future Research Directions

How health conditions may influence trust is a topic largely unexplored in sociopsychological research. Our study provides some new evidence suggesting that this direction of influence in the health–trust relationship is plausible. Making connections to the possible underlying psychophysiological mechanisms is helpful when formulating conceptual models and interpreting the statistical results. Our study demonstrates that the information processing approach of mindsponge-based reasoning [27] and BMF analytics [29] are useful in this type of investigation. We suggest that interdisciplinary psychosocial studies that utilize existing evidence from natural sciences can help expand new boundaries in trust research. Additionally, policymakers should be aware not only that trust of the right type in the right context can promote good health but also that good public health may promote higher social trust.

### 4.4. Limitations

The study has some limitations. Firstly, the possible psychophysiological pathways presented in the paper need more direct evidence from neurobiological experiments on humans to be confirmed. Secondly, the used dataset is from a relatively low-trust social environment. Further studies using data from high-trust societies can crosscheck and compare the patterns found in this study. Thirdly, the aspect of culture was not examined. Future research efforts may incorporate cultural values since they may play an essential role in shaping a society’s collective mindset and trust patterns. Fourthly, sex and age can be potential physiological factors that may affect the mental processes involving trust, which can be good targets of investigation using the information-processing approach. This study is an early attempt to explore how human physiology can affect the cognitive processes of trust. We hope that this can be a potential and exciting direction of trust research in the near future.

## Figures and Tables

**Figure 1 healthcare-11-02373-f001:**
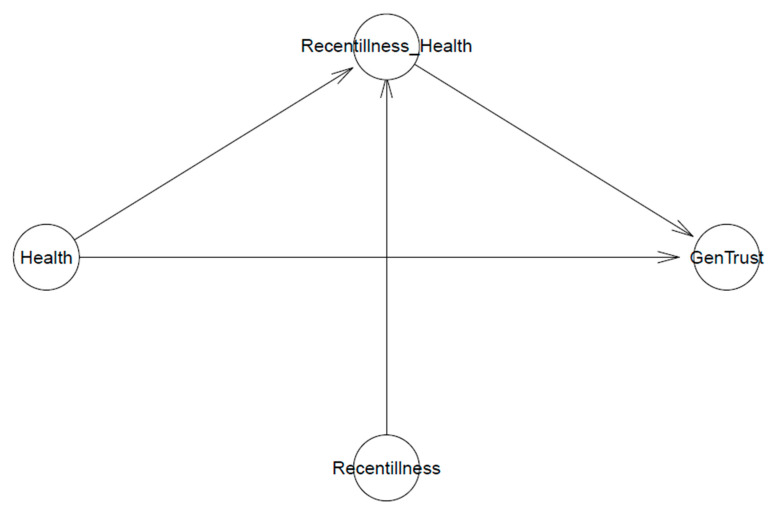
Model 1’s logical network.

**Figure 2 healthcare-11-02373-f002:**
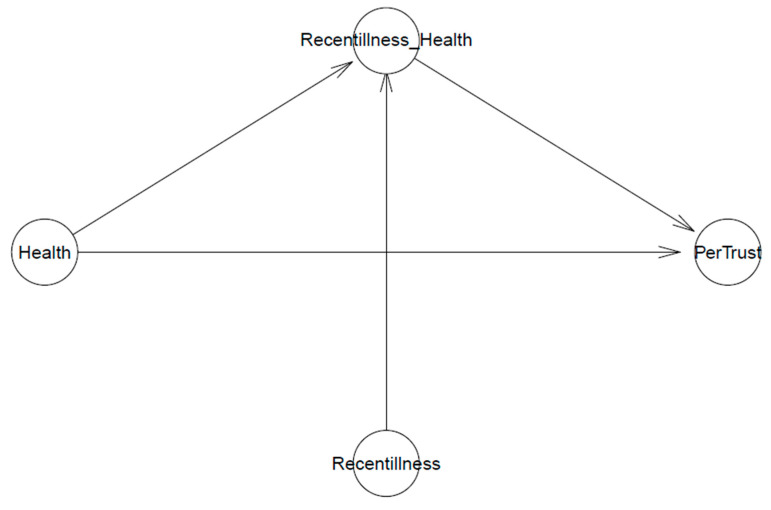
Model 2’s logical network.

**Figure 3 healthcare-11-02373-f003:**
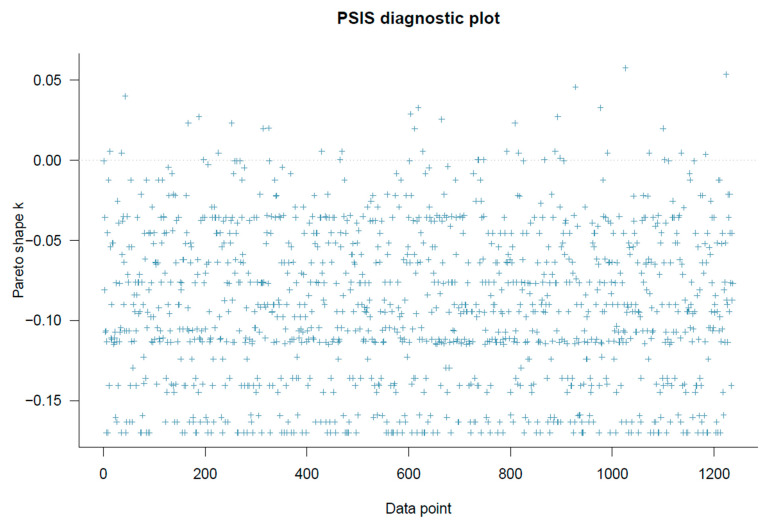
Model 1’s PSIS-LOO diagnostic plot.

**Figure 4 healthcare-11-02373-f004:**
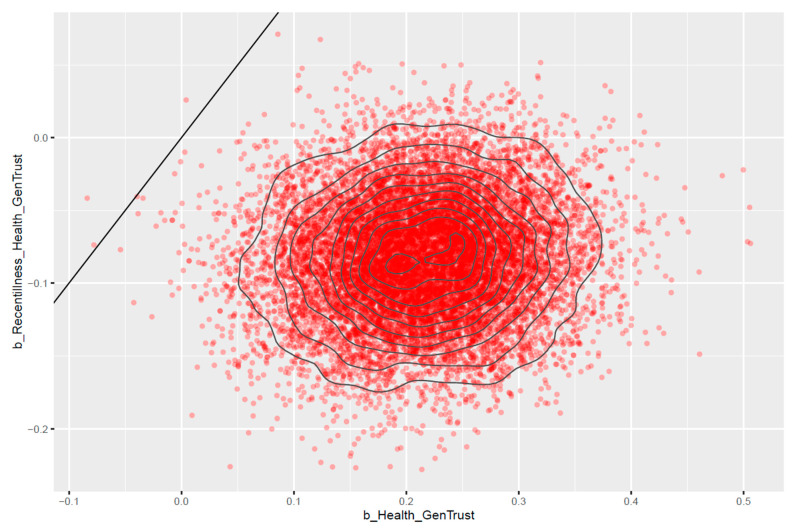
Pairwise distribution plot for model 1’s *Health* and *Recentillness ∗ Health*.

**Figure 5 healthcare-11-02373-f005:**
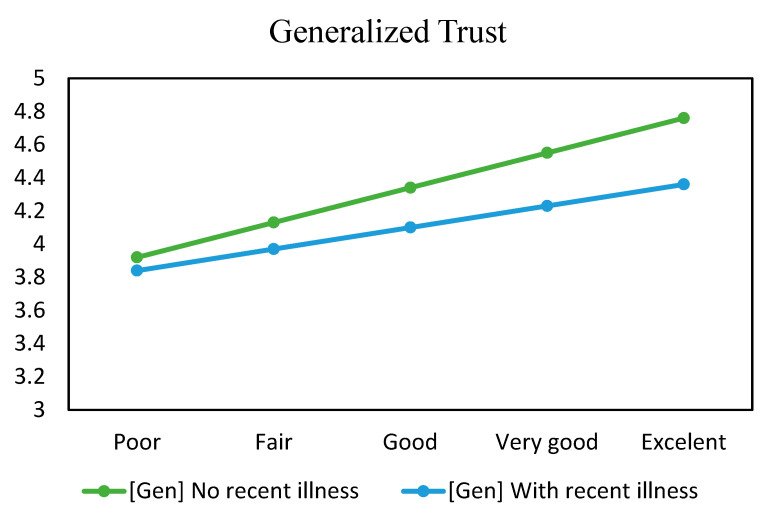
Estimated generalized trust values based on health status and recent illness experience.

**Figure 6 healthcare-11-02373-f006:**
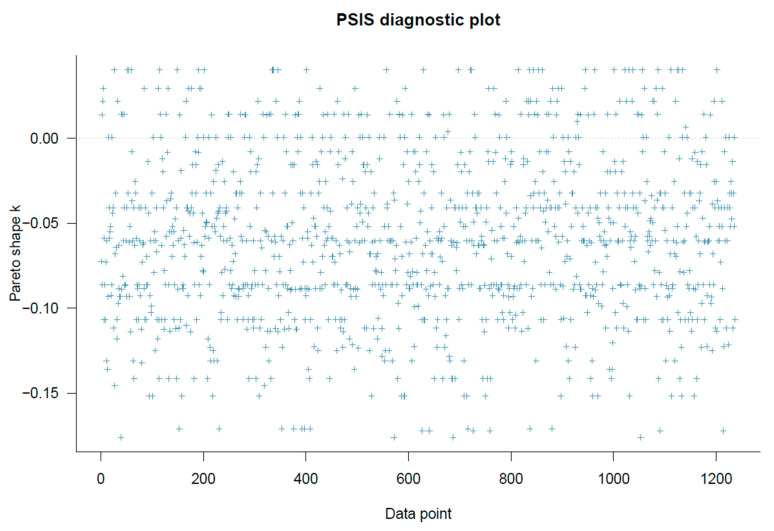
Model 2’s PSIS-LOO diagnostic plot.

**Figure 7 healthcare-11-02373-f007:**
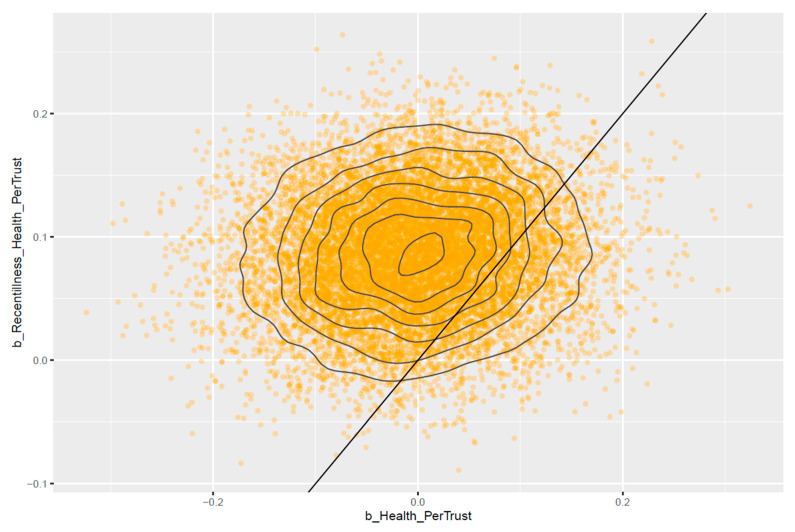
Pairwise distribution plot for Model 2’s *Health* and *Recentillness ∗ Health*.

**Figure 8 healthcare-11-02373-f008:**
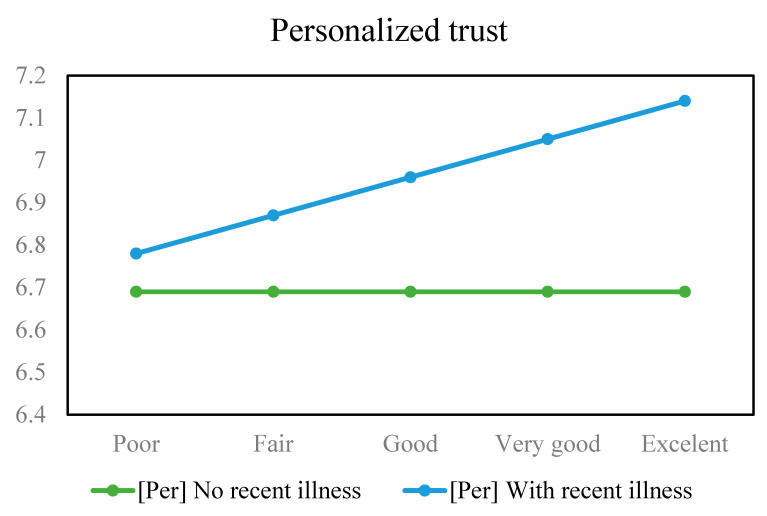
Estimated personalized trust values based on health status and recent illness experience.

**Table 1 healthcare-11-02373-t001:** Variable description.

Variable	Meaning	Type of Variable	Value
*GenTrust*	The respondents’ degree of generalized trust	Ordinal	From 0 (not at all) to 10 (completely)
*PerTrust*	The respondents’ degree of personalized trust	Ordinal	From 0 (not at all) to 10 (completely)
*Health*	The respondent’s general health status	Ordinal	From 1 (poor) to 5 (excellent)
*Recentillness*	Whether the respondent experienced physical illness in the past 30 days	Binary	0 (no) or 1 (yes)

**Table 2 healthcare-11-02373-t002:** Model 1’s estimated posteriors.

Parameters	Mean	SD	*n_eff*	*Rhat*
*Constant*	3.71	0.33	4977	1
*Health*	0.21	0.07	5062	1
*Recentillness ∗ Health*	−0.08	0.04	7349	1

**Table 3 healthcare-11-02373-t003:** Model 2’s estimated posteriors.

Parameters	Mean	SD	*n_eff*	*Rhat*
*Constant*	6.69	0.39	5270	1
*Health*	0.00	0.08	5410	1
*Recentillness ∗ Health*	0.09	0.05	8348	1

## Data Availability

All data and code snippets of this study were deposited at an Open Science Framework server (https://osf.io/czd9t/) (accessed on 13 March 2023).

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
