# Peer review of "Trust Is for the Strong: How Health Status May Influence Generalized and Personalized Trust"

_healthcare, 2023, doi:10.3390/healthcare11172373_

Round 1

Reviewer 1 Report

TOPIC:  This is a valuable initiative and well set out with the TRUST-> HEALTH relationship and relevant resources. Exploring the HEALTH -> TRUST  direction is key and may provide some insights not only into interpersonal relationships but also into trust in healthcare providers AND treatments. 

PERSONALIZED TRUST:  You don't define personalized trust until line 56 in the paper. Would you be able to find a way you can work its working definition into a sentence that is already in the abstract for initial context? Then generalized trust can also be more readily discerned by contrast. 

RESEARCH QUESTIONS:  Very much resonated with the 4 key research questions and their interrelationships, and appreciated that you were very clear on the " possible direction of health influencing trust" perspective. Interesting selection of analytics, but justified. 

MINDSPONGE DEFINITION:  When outlining mindsponge: this needs more explanation as it seems slightly jargoned: "The mind, according to the mindsponge theory, is a collection-cum-processor for information, which includes biological and social systems of various complexity levels". This does not provide sufficient clarity for that platform. 

The further explanation of "A mindsponge information process has the following characteristics: .." where you outline 6 elements, helps [FORMAT:  Indent the 6 elements for readability]

NEUROPLASTICITY: Are we already meant to know this term in this domain? A brief definition would help those who don't. 

FUNCTION:  The Bayes' contingency function in line 143 is clearly set out and explained theoretically. Should the equations be numbered for this publication format?

RECENT ILLNESS:  This needs more clarity:  “Now thinking about your physical health, which includes physical illness and injury, for how many days during the past 30 days was your physical health not good?” How is this operationally defined? What guidance do people have for "not good"? Missing work, affecting daily function to a given degree ... Is this guidance provided? 

TRUST METRIC(S):   You noted:  The survey follows international guidelines and includes three questions about health adapted from the Centers for Disease Control and Prevention. The interpersonal and institutional trust component was based on OECD guidelines to measure trust. Was there a link to this tool? If not provide one. If so, I missed it. HOW did you choose this tool? What is its outlined validity or track record? 

MODELS' LOGICAL NETWORKS and EQUATIONS/VARIABLES: Figures 1 & 2 look nearly identical. (1) Emphasize that one includes general and the other depicts personalized in the Figure, not just in the body of the text. (2) refer to each symbol directly from the functions to help the reader track more easily with the meaning of each variable.  For example, after the equation set, write

"Where: ..."

... and then outline each variable.

FIGURES 3, 4, 6 & 7's captions can/should include more information about (1) what is being seen and (2) what it actually shows/meaning of findings. You do expand upon this in the body of the text, but help the Figures stand with more independence. 

FIGURES 5 & 6:  These are helpful and informative plots. Could you convert the x-axis to Likert-type ordinal values and indicate an R^2 value for these relationships, although they are characterized as "estimates"? Very interesting findings!

SUBSECTIONS:  If permitted, Subsections - particularly in the Introduction and Discussion sections would provide more clarity and resolution for (1) the foundation and (2) the findings, respectively. 

LIMITATIONS: Good that you recognized these, especially in the area of neuro-bio experimentation and the baseline trust level. 

FUTURE WORK: It would be great to expand on this from your deep work. What more can be done/asked in addition to the cultural element?

REFERENCES:  A strong set of resources 

OVERALL:  Great work using tying together psychological elements with health and algorithmic application of the effects in both directions. 

Reviewer 2 Report

This paper seems to be a post-hoc analysis of four variables outlined in Table 1: Variable description. The general “story” of the research should be explainable using these variables. BUT, the story seems to be “we looked for an association using a new statistical model.” After reading the article, I gained no insight into the mechanism of how these variables constitute a believable model. Libraries have been written about the “mind and society” (Line 181) The introduction and methods need to articulate how this paper furthers our understanding of that issue.

https://www.sciencedirect.com/science/article/pii/S2215016122001881

This seems like a significant flaw because Mindsponge methodology requires research questions (of significance) and construction of a hypothetical model.

This hypothetical model is described in 2.2.2 Model Formulation—I would have expected this to appear much earlier in the article.

Lines 243-250 seem to indicate Four questions were used to measure the variables. Yet the authors state they used OCED guidelines (line 225).

https://www.oecd-ilibrary.org/docserver/9789264278219-en.pdf?expires=1687538420&id=id&accname=guest&checksum=DB461FD9AA56F60E7E03672FFB654F1C

The introduction was confusing at best. A simplified explanation that explains why you chose your model would improve the manuscript. 

The English was difficult to follow. Copyediting is suggested.
